# Impact of Lifestyle Behaviors on Postprandial Hyperglycemia during Continuous Glucose Monitoring in Adult Males with Overweight/Obesity but without Diabetes

**DOI:** 10.3390/nu13093092

**Published:** 2021-09-02

**Authors:** Ichiro Kishimoto, Akio Ohashi

**Affiliations:** 1Department of Endocrinology and Diabetes, Toyooka Public Hospital, 1094 Tobera, Toyooka 668-8501, Japan; 2Environment and Total Quality Management Division, NEC Corporation, 5-7-1 Minato-ku, Shiba, Tokyo 108-0014, Japan; ohashi_akio@nec.com

**Keywords:** habitual snacking, hyperglycemia, disposition index, physical inactivity, continuous glucose monitoring

## Abstract

Data regarding hyperglycemia-related factors were scarce in people without diabetes. Fifty males (age 50–65 years) with overweight/obesity but without diagnosis of diabetes were recruited. After excluding participants with the 2 h plasma glucose value during a 75 g oral glucose tolerance test ≥200 mg/dL, continuous glucose monitoring (CGM) was performed for 6 days. Subjects with ≥1800 CGM readings were included (*n* = 36). The CGM indices of hyperglycemia were significantly associated with disposition index and snacking frequency. In receiver-operating characteristic analysis for predicting the maximal CGM glucose ≥200 mg/dL, the area under curves of disposition index, snacking frequency, and minimal daily step counts during the study were 0.69, 0.63, and 0.68, whereas the cutoff values were 1.57, once daily, and 2499 steps, respectively. After adjustments, the lower disposition index (≤1.57), higher snacking frequency (≥1 per day), and lower minimal step (≤2499 steps per day) categories conferred 14.5, 14.5, and 6.6-fold increased probabilities for having the maximum level ≥ 200 mg/dL, respectively. In addition, the snacking habits were significantly associated with insulin resistance and compensatory hyperinsulinemia. In conclusion, in middle aged males with overweight/obesity but without diabetes, snacking and physical inactivity serve as the major drivers of postprandial hyperglycemia independently of β-cell function.

## 1. Introduction

Obesity and insulin resistance are major risk factors for type 2 diabetes [1]. When the adaptive response of β-cells to increased insulin resistance is insufficient, hyperglycemia develops [2], which, along with other cytotoxic factors, further diminishes β-cell mass and function possibly through β-cell apoptosis [3]. Since, by the time that type 2 diabetes is diagnosed, 50–80% of β-cell function is lost [1,4] and there is a significant reduction (24–65%) in β-cell mass [3,5], efforts for the prevention of β-cell dysfunction should be undertaken before the onset of diabetes. In fact, it is suggested that the onset of β-cell dysfunction may commence many years before the diagnosis of the disease [6]. However, the time point at which this abnormality begins and the factors that may be responsible for this pathological and functional change have not been elucidated [7].

Hyperglycemia causes inflammation, which consequently results in β-cell apoptosis. When the extent of apoptosis due to repeated hyperglycemia exceeds the extent of regeneration of the pancreatic β-cell mass, the overall insulin secretion in response to a glycemic load is reduced, which results in more hyperglycemia [8]. Therefore, hyperglycemia is both a causative factor and an early marker for β-cell dysfunction before the onset of diabetes. Thus, understanding the extent of hyperglycemia in individuals without diabetes under real-life conditions is of great significance for preventive strategies and public health initiatives. However, in individuals without diabetes, exposure to elevated glucose levels remains underappreciated [9], and there are limited data on the association between a β-cell dysfunction measure or lifestyle factors and glucose excursions.

Age, male gender, and overweight/obesity are the major risk factors for developing type 2 diabetes. Recently, we have examined postprandial glucose in middle-aged men with overweight/obesity, most of which were normal on the glucose challenge test, and demonstrated that a substantial proportion exhibited elevated glucose levels above the recommended target for diabetes management [10]. Therefore, it is suggested that caution must be exercised in order to prevent postprandial hyperglycemia, even without diabetes. In this study, in order to determine the factors associated with hyperglycemia in people without diabetes, continuous glucose monitoring (CGM) was performed in middle-aged men with overweight/obesity but without diabetes, and the associations between the factors related to insulin secretion/sensitivity or to lifestyles and the glycemic profiles were examined in daily life. Among lifestyle behaviors, we particularly focused on diet and exercise, the two major modifiable factors for hyperglycemia.

## 2. Methods

### 2.1. Study Participants

A total of 50 (body mass index [BMI] ≥25 kg/m^2^) middle-aged (age 50–65 years) male participants, who had undergone a medical check-up within 1 year and have no documented dysglycemia (previous diagnosis of diabetes or HbA1c ≥6.5% (48 mmol/mol) or FPG ≥126 mg/dL (7 mmol/L) in the preceding 1 year), were recruited into this study. After oral glucose tolerance tests (OGTT), persons with a 2 h post challenge glucose value of less than 200 mg/dL (11.1 mmol/L) were selected and asked to wear CGM devices. Individuals whose CGM recording data were obtained for 6 days, with more than 1800 CGM readings, were chosen for inclusion in the analysis (*n* = 36). For the analyses including step counts, a participant who did not record the steps during the study was excluded.

### 2.2. Assessing Glycemia

Glucose levels were assessed by using four different methods: (1) CGM was performed using iPro™2 Professional CGM (Medtronic, Minneapolis, MN, USA); (2) participants created a seven-point self-monitored blood glucose profile (preprandial, 1~2 h postprandial, and pre-bedtime) by using a glucometer (Glutest Neo Alpha; Sanwa Kagaku Kenkyusho Co, Osaka, Japan) during the days of CGM; (3) in the overnight fasting state, the venous blood was drawn from the cubital vein and the fasting plasma glucose (FPG), serum glycated hemoglobin (HbA1c), and plasma 1,5-anhydroglucitol (AG) were measured using standard laboratory procedures; (4) a 2 h, 75 g OGTT was performed after overnight fasting (≥12 h). The cutoff 2 h plasma glucose levels of 140 mg/dL (7.8 mmol/L) were used to diagnose normal (NGT) and impaired glucose tolerance (IGT).

### 2.3. Insulin Secretion/Resistance Indices

Based on simultaneous measurements of blood glucose and insulin concentrations under fasting conditions or during the OGTT, indirect indices of insulin secretion or insulin resistance were determined. The homeostatic model assessment (HOMA)-β and the quantitative insulin sensitivity check index (QUICKI) were determined by the following formulae: fasting insulin × 360/(fasting glucose − 63) and 1/(log[fasting insulin] + log[fasting glucose]), respectively [11]. HOMA-IR, the insulinogenic index, Matsuda Index [12], and disposition index (the product of insulinogenic index and Matsuda Index) were calculated online at http://mmatsuda.diabetes-smc.jp/MIndex.html (accessed on 30 August 2021). Formulae for HOMA-IR, insulinogenic index, and Matsuda were (OGTT PG 0 × OGTT IRI 0)/405, (OGTT IRI 30 − OGTT IRI 0)/(OGTT PG 30 − OGTT PG 0), and 10,000/SQRT((OGTT PG 0 × OGTT IRI 0) × ((OGTT PG 0 + OGTT PG 30 × 2 + OGTT PG 60 × 3 + OGTT PG 120 × 2)/8 × (OGTT IRI 0 + OGTT IRI 30 × 2 + OGTT IRI 60 × 3 + OGTT IRI 120 × 2)/8)), respectively. In order to calculate the insulin secretion-sensitivity index (ISSI)-2, the area under the curve (AUC) of the insulin (μU/mL) and glucose (mg/dL) levels was determined by the trapezoidal rule during OGTT, and the ratio of the AUC for insulin to that of glucose was multiplied by the Matsuda Index.

### 2.4. Study Protocol

On the first day, anthropometric data were obtained, and a venous blood sample from the cubital vein was obtained to measure FPG, HbA1c, and 1,5-AG. Then, the 75 g OGTT was performed, and a CGM device was attached to the abdomen. The iPro™2 recorder and Enlite sensor were worn for 6 days. The participants were instructed to calibrate the sensor according to the manufacturer’s specifications four times throughout the day. Consumptions of foods and drinks, wake-up/sleep time, present illness, medications, supplements, subjective symptoms (dyspepsia and bowel movement, etc.), and emotional or physical stress (hectic schedule and interpersonal relations, etc.) were self-reported using the questionnaire on the first day. During the study period, participants were asked to keep daily logs of food intake and exercise and to take photographs of every meal/snack/drink content with date and time stamp with a digital camera (COOLPIX, Nikon, Tokyo, Japan). Participants were also instructed to wear a smartwatch-type activity tracker (PULSENSE, EPSON, Tokyo, Japan) all day, except at bath time.

### 2.5. Determination of Snacking and Drinking Frequencies

Eating or drinking occasions were identified by the participants themselves. Diet-related data were used to calculate the average number of occasions on which the participant consumed food or drink items per week (for self-reported habits) or per day (averaged during the study). Where more than two occasions occurred in the same day for self-reported habits or in the same 15 min period for frequencies during the study, all events were taken as a single occasion. During the study, the food or drink groups and times of the ingestion were evaluated using the lifestyle logs and the photographs. Participants were asked to identify each food as breakfast, lunch, or dinner for calibration of the CGM sensor, and this information was used to identify meals for the purpose of the analysis. In the present study, we defined snacking as the ingestion of any caloric food other than meals, and drinking as the ingestion of alcohol from the self-reported questionnaires (for habits) or from the lifestyle logs (during the study). The average numbers of occasions on which the participant consumed food or drink items per week (for habits) and per day (during the study) were calculated.

### 2.6. Assessment of Physical Activity

In the present study, maximal, average, and minimal daily walking steps during the study were assessed by using smartwatch-type activity trackers and included in the analysis.

### 2.7. Data Analysis for CGM

As indicators of dysglycemia, the mean glucose level, maximal glucose level, standard deviation (SD), percent coefficient variability (CV), time above range (TAR; the percentage of time when the CGM-recorded value exceeds the selected glucose thresholds), and percentage of postprandial peaks above the selected glucose thresholds were calculated by using all CGM glucose data obtained during the study. The glucose concentrations corresponding to the cutoff points that were proposed as clinical targets [13] were used as the thresholds.

### 2.8. Statistical Analysis

Baseline data are expressed as median and interquartile range (IQR) for all participants or stratified by the snacking habits category. For categorical data, Pearson’s chi-square test was used to evaluate how likely it was that any observed difference between the sets increased coincidentally. In order to measure the strength and direction of association between glucose/insulin-related parameters and CGM metrics, the Spearman’s rank-order correlation coefficients (ρ) were calculated. The Mann–Whitney U test was used for the comparison of variables based on the categorical data. The participants were categorized by the frequency of consumption of meals (three times per day or otherwise) or snacking (≥once a day or less), by other diet-related parameters (≥once during the study period or none), or by the cutoff values obtained from the receiver-operating characteristic (ROC) curves. Participants were also stratified by if they have drinking or snacking habits, which was identified by responses in a self-reported survey questionnaire. For the lifestyle factors related to exercise, participants were divided by the median into subgroups with “higher” and “lower” lifestyle values.

Multiple linear regression was fitted for the maximal CGM sensor glucose value, with the disposition index, snacking times per day, and the minimal step category as the predictive factors. The normality of residuals was validated by the Shapiro–Wilk test (W = 0.9796; *p* = 0.7476). The variance inflation factor (VIF) calculated for each predictor was <2.5, indicating that multicollinearity could be safely ignored. In addition, a multiple logistic regression model was constructed to examine the association between the maximal CGM glucose ≥200 mg/dL category and independent variables, including the disposition index category (cutoff, 1.57), the snacking times per day category (cutoff, once a day), and the minimal step category (cutoff, 2499 steps per day). In order to integrate a two-level categorical variable into the regression models, a dummy variable with two values was created by assigning 1 for the objective category and −1 for the control category. The odds ratios of having a maximal CGM glucose level ≥ 200 mg/dL were calculated via the maximum likelihood method in the logistic regression models. Since type I error rates were similar for 5–9 and 10–16 events per predictor variable [14], we conducted fitting with all three binary variables in addition to one and two variables.

The ROC curve analysis was applied to measure the diagnostic accuracy of the disposition index, snacking frequency, or minimal step count for predicting the maximal CGM glucose level ≥200 mg/dL. For a measure of goodness of fit for binary outcomes in a logistic regression model, the AUC was computed. The cutoff point was determined via the Youden index to maximize the overall accuracy of the classification rate and assign equal weight to the sensitivity and the specificity.

Based on the assumption that the effect sizes of lifestyle behaviors are large, sample sizes were calculated as 34 for Spearman’s rank-order test (two tails; effect size = 0.5; α error = 0.05; power = 0.9) and 42 for Mann–Whitney U test (one tail; effect size = 0.8; α error = 0.05; power = 0.8).

Since the present study aims to test the prespecified hypotheses that certain specific lifestyle factors are related to the frequency of hidden hyperglycemia, the significant differences found in the study are not from the post hoc comparison but from the a priori planned comparisons. Therefore, we did not perform the correction for multiple tests.

Statistical significance was defined as a *p* value of <0.05.

### 2.9. Ethics Statement

The study was performed in accordance with the principles established by the Helsinki Declaration and approved by the institutional review board of Toyooka Public Hospital (#146; 3 October 2017) and the Japan Conference of Clinical Research review board (JCCR#3-132; 21 October 2016). Written informed consent was obtained from all participants prior to study enrollment.

## 3. Results

The characteristics of all study participants were reported previously [10]. In brief, the median (IQR) age and BMI were 54 (52–58) and 27.9 (26.5–29.4), respectively. Median (IQR) HbA1c was 5.4 (5.2–5.6)%, equaling 35.5 (33.3–37.7) mmol/mol, whereas median (IQR) 1,5-AG level was 19.7 (15.3–24.1) µg/mL. Although the β-cell function estimated by the HOMA-β was well preserved (median 101, IQR 71–154), approximately a quarter of the study population had the insulinogenic index of <0.4 (the median insulinogenic index was 0.61 (IQR 0.38–1.24)). The results of the 75 g OGTT revealed that 73% had NGT, whereas 27% had IGT. The medians (IQR) at 1 h and at 2 h post-challenge glucose levels during OGTT were 176 (150–194) mg/dL and 112 (96–140) mg/dL, respectively.

The CGM results, for which the median total count was 1964 (163.7 h), showed that the median maximal CGM sensor glucose level was 193 (IQR, 173–219) mg/dL. Approximately half (47%) of the participants had CGM-recorded glucose levels of ≥200 mg/dL at least once. In participants with the maximal CGM glucose level ≥200 mg/dL, the frequencies of having more than 20% of postprandial CGM peaks ≥180 mg/dL and having more than 40% of postprandial CGM peaks ≥140 mg/dL were significantly higher than in those with the maximal CGM glucose level <200 mg/dL (69% vs. 10%, *p* = 0.0008; and 91% vs. 36%, *p* = 0.0012, respectively). The median (IQR) TARs higher than 140 and 200 mg/dL were 10.4 (4.3–15.8)% and 0.0 (0.0–0.73) %, respectively, while the median (IQR) percentages of ≥140 and ≥200 mg/dL postprandial peaks were 57.5 (25.4–75.5)% and 0 (0–6.4)%, respectively. Logging of foods by taking photographs with digital cameras revealed that the median (IQR) frequency of snacking was 0.43 (0.04–0.86) times per day. The self-reported diet questionnaire revealed that 33.3% of the participants eat snacks habitually (answered “yes” to whether they have snacking habits). There was no significant correlation between the snacking frequency during the study (average times per day) and the self-reported frequency of the habitual snacking (snacking days per week) (Spearman’s ρ = 0.2, *p* = 0.234). The pedometer recordings of the number of steps revealed that the participants walked approximately 7000 steps per day on average.

When the associations of anthropometric, glucose/insulin-related, and diet/exercise-related parameters with the CGM metrics were examined, the HbA1c and disposition index were significantly associated with all examined CGM parameters for hyperglycemia positively and negatively, respectively (Table 1). The preload fasting glucose level was significantly associated with TAR > 140 mg/dL, whereas the 1 h post-challenge glucose level and Matsuda index were significantly associated with the maximal CGM glucose level, TAR >140 mg/dL, and the frequency of the ≥140 mg/dL postprandial CGM glucose peak (Table 1). Among the lifestyle-related indices, the self-reported snacking habits (days per week) were significantly associated with the maximal CGM glucose level, TAR >200 mg/dL, and the percentages of the ≥140 mg/dL and the ≥200 mg/dL postprandial CGM glucose peaks, while the snacking frequency during the study (times per day) was positively associated with the frequency of the ≥140 mg/dL postprandial CGM glucose peak (Table 2).

The associations between the dichotomous categories of the lifestyle-related indices and the CGM metrics were next examined. In participants who consumed at least one snack per day on average during the study, the maximal CGM glucose level, TARs, and the frequencies of hyperglycemic peaks were significantly higher than in those who consumed less than one snack per day (Table 3). In addition, when the participants were categorized by the median of daily step counts, the maximal CGM glucose level, TARs, and the frequencies of hyperglycemic peaks were significantly higher in the lower minimal step category (less than the median, at 2681 steps per day) than in the higher minimal step category (≥2681 steps per day), while there were no significant differences in the hyperglycemic indices between the categories of average or maximal walking steps (Table 3).

In the ROC analysis for sensitivity and specificity in detecting the maximal CGM glucose level ≥200 mg/dL, the AUC and the cutoff value of snacking frequency were 0.63 and 1.0 times per day (sensitivity 41.2% and selectivity 94.7%), respectively, whereas the AUC and the cutoff of the disposition index were 0.69 and 1.57 (sensitivity 47.1% and selectivity 94.7%), and those of minimal step count were 0.68 and 2499 (sensitivity 76.5% and selectivity 73.7%), respectively.

When the participants were categorized by the cutoff values, univariate or bivariate analysis showed that the dichotomized explanatory variables were significantly correlated with the maximal CGM glucose level ≥200 mg/dL (Table 4). In the model, including all the categories of the disposition index, the snacking frequency, and the minimal step count as predictor variables, the three variables were independently associated with the postprandial hyperglycemia (Table 4). The area under the ROC curve of the multivariate model was 0.863 (*p* = 0.0003). After adjustment, participants with a lower disposition index (≤1.57), higher snacking frequency (≥1 a day), and lower minimal step count (≤2499) had a 14.5-fold, 14.5-fold, and 6.6-fold increased probability for having the maximal CGM glucose level ≥200 mg/dL, respectively, compared to the participants in the opposite category (Table 4).

As shown in Table 5, in participants with the self-reported snacking habits, the CGM maximal glucose, TAR, and the frequencies of postprandial hyperglycemia were significantly higher than in those without the habits. In addition, the HOMA-β and fasting/post-challenge insulin levels were higher, whereas the Matsuda Index was lower, indicating insulin resistance with compensatory hyperinsulinemia in participants with snacking habits.

## 4. Discussion

In the present study, we performed CGM in adult male with overweight/obesity but without diabetes and found that, even in non-diabetic participants, a substantial proportion of participants exhibited elevated sensor glucose levels above the recommended target for diabetes management. The lower disposition index category (≤1.57), the higher snacking frequency category (≥1 a day), and the lower minimal step count categories (≤2499) were independently associated with the maximal CGM glucose level ≥200 mg/dL, indicating that snacking and physical inactivity result in hyperglycemia independently of the insulin secretion.

In individuals without diabetes, it is reported that the plasma glucose concentrations peak approximately 60 min after the start of a meal, rarely exceed 140 mg/dL, and return to the preprandial levels within 2–3 h [15,16]. However, in this study, among men with overweight/obesity but without diabetes, 100% and 47% of the participants exhibited the maximal CGM glucose levels ≥140 mg/dL and ≥200 mg/dL, respectively, indicating that postprandial hyperglycemia prevails among non-diabetes. Since there were significant associations between the 1 h post-challenge glucose level and the CGM indices of hyperglycemia (Table 2), a diagnosis of dysglycemia in individuals without diabetes should be made on the basis of the 1 h and not of the 2 h post-challenge glucose level.

In this study, snacking is associated closely with the indices of hyperglycemia. As shown in Table 3 and Table 4, snacking at least once a day is closely associated with indices of hyperglycemia. Interestingly, during the study period, self-reported snacking habits (and not snacking frequency) were significantly associated with indices of insulin resistance such as HOMA-IR, the Matsuda Index, or QUICK. After adjustment for age and BMI, the odds ratio of having a value in the lower half of the Matsuda Index was 7.9 times higher in participants with snacking habits than in those without these habits. Therefore, while snacking results in postprandial hyperglycemia, snacking habits, through the continued consumption of unhealthy snacks, produce hepatic and peripheral insulin resistance, which further worsens the hyperglycemia. Since the strength of habits is the most important predictor in explaining unhealthy snack intake [17] and since poor dietary habits established during childhood could persist into adulthood, which increases the risk of developing obesity and obesity-related complications such as type 2 diabetes [18], it is important to encourage the development of healthy eating habits as early as possible so that they can last a lifetime.

In this study, the lower minimal step category was significantly associated with higher CGM metrics for mean and postprandial hyperglycemia, whereas intergroup differences among categories of average or maximal steps did not reach statistical significance. Minimal steps during the study less than 2499 per day effectively predicts a maximal CGM glucose level of ≥200 mg/dL. As prolonged sitting and less physical activity are integrated into modern lifestyles across settings such as transportation, the workplace, and the home, a sedentary lifestyle has lately received focus as a major mortality risk factor [19,20], independent of physical activity [21]. A randomized crossover study showed that interrupting the sitting duration with standing and light-intensity walking effectively improved daily glucose levels and insulin sensitivity in type 2 diabetes [22]. These results suggest that the lower number of minimal steps could be a surrogate indicator of a sedentary lifestyle and attempts to increase minimal step counts in daily living could be beneficial for preventing dysglycemia.

Limitation of the study includes the following: (1) The sample size calculation was based on the assumption that the effect size of snacking is large. The study may, therefore, not have the statistical power to expose possible smaller effects of other lifestyle factors (a type II error). The (2) present study focused on middle-aged male with overweight/obesity, one of the highest groups for developing type 2 diabetes. The association of lifestyle factors and hyperglycemia in other population should be determined elsewhere. Since (3) habitual lifestyle behaviors are self-reported, participants might overestimate or underestimate their eating, snacking, or drinking habits. During the study, we evaluated food items by photographs where a reporting bias may also exist and (4) since, for dietary behaviors, only the occasions were taken account, the present study does not tell the effects of the intensity of snacking or drinking other than occasions. In addition, no data regarding the intensity of physical activity other than step counts exist.

## 5. Conclusions

In adult male with overweight/obesity but without diabetes, higher snacking frequency and lower walking steps serve as the major drivers of postprandial hyperglycemia, independently of β-cell function. Since the snacking habits are associated with insulin resistance, which further worsens the hyperglycemia, detecting hyperglycemia as early as possible and breaking unhealthy habits could prevent detrimental glucose surges in daily living and mitigate dysfunction of β-cells.

## Figures and Tables

**Table 1 nutrients-13-03092-t001:** Spearman’s rank-order correlation coefficients (ρ) of significant associations between biochemical indices and CGM metrics.

Parameters	CGM Max	TAR > 140	TAR > 200	% of ≥140 Peak per Meal	% of ≥200 Peak per Meal
HbA1c, %	0.52	0.72	0.39	0.65	0.37
1,5-AG, μg/mL	−0.33				
OGTT PG 0, mg/dL		0.34			
OGTT PG 30, mg/dL	0.40	0.35			
OGTT PG 60, mg/dL	0.33	0.43		0.40	
OGTT PG 120, mg/dL					
OGTT IRI 0, μU/mL		0.38			
OGTT IRI 30, μU/mL					
OGTT IRI 60, μU/mL		0.35			
OGTT IRI 120, μU/mL	0.42	0.49	0.35	0.40	
Insulinogenic index					
HOMA-β					
HOMA-IR		0.41			
Matsuda index	−0.33	−0.52		−0.39	
Disposition index	−0.48	−0.52	−0.40	−0.44	−0.38
QUICKI		−0.45		−0.34	

To describe the strength of the association between the two variables, Spearman’s rank-order correlation coefficients were calculated. Only correlations with statistical significance (*p* < 0.05) were shown. 1,5-AG, 1,5-anhydroglucitol; HOMA, homeostatic model assessment; QUICKI, quantitative insulin sensitivity check index; OGTT, 75 g oral glucose tolerance test; PG 0, 30, 60, and 120 pre-load and 30 min, 60 min, and 120 min post-load plasma glucose levels, respectively; IRI 0, 30, 60, and 120 pre-load and 30 min, 60 min, and 120 min post-load serum insulin levels, respectively; CGM max, the maximal sensor glucose level during CGM; TAR, time above range; TAR > 140 and 200, the percentages of time above sensor glucose 140 and 200 mg/dL, respectively; % of ≥140 and 200 peak per meal, proportions of postprandial hyperglycemia equal to or exceed 140 and 200 mg/dL, respectively.

**Table 2 nutrients-13-03092-t002:** Spearman’s rank-order correlation coefficients (ρ) of significant associations between lifestyle-related indices and CGM metrics.

Parameters	CGM Max	TAR > 140	TAR > 200	% of ≥140 Peak per Meal	% of ≥200 Peak per Meal
Skip breakfast (4–10 a.m.), % of days					0.33
Late dinner (10 p.m.), % of days					
Drinking habits, days per week		−0.35			
Snacking habits, days per week	0.33		0.41	0.42	0.39
Drinking frequency, times per day			−0.34		−0.38
Snacking frequency, times per day				0.39	
Average walking step counts, steps per day					
Maximal walking step counts, steps per day					
Minimal walking step counts, steps per day					

To describe the strength of the association between the two variables, Spearman’s rank-order correlation coefficients were calculated. Only correlations with statistical significance (*p* < 0.05) were shown. CGM max, the maximal sensor glucose level during CGM; TAR, time above range; TAR > 140 and 200, the percentages of time above sensor glucose 140 and 200 mg/dL, respectively; % of ≥140 and 200 peak per meal, proportions of postprandial hyperglycemia equal to or exceed 140 and 200 mg/dL, respectively. In the present study, we defined snacking as ingestion of any food other than meals and drinking as ingestion of alcohol.

**Table 3 nutrients-13-03092-t003:** Median (IQR) of CGM metrics according to the lifestyle-related factor categories.

Parameters	Category	*n*	CGM Max	TAR > 140	TAR > 200	% of ≥140 Peak per Meal	% of ≥200 Peak per Meal
Skip breakfast (4–10 a.m.)	≥once during the study	17	207 (175–228)	11.0 (5.3–19.9)	0.4 (0–0.92)	71.4 (34.3–83.3)	4.76 (0–10.3)
	none	19	184 (172–213)	9.4 (2.7–15.5)	0 (0–0.46)	42.9 (23.8–66.7)	0 (0–4.76)
	*p* value		0.241	0.384	0.14	0.188	0.093
Late dinner (10 p.m.)	≥once during the study	12	206 (190–234)	11.5 (8.5–17.8)	0.41 (0–0.73)	64.3 (52.2–75.5)	4.76 (0–8.81)
	none	24	184 (168–215	8.5 (2.4–15.8)	0 (0–0.81)	43.9 (23.6–76.8)	0 (0–5.42)
	*p* value		0.07	0.159	0.21	0.383	0.275
Drinking habits	yes	11	180 (172–187)	5.2 (2.3–11.4)	0 (0–0)	40.9 (23.8–63.2)	0 (0–0)
	no	25	205 (176–220)	11.7 (5.3–17.7)	0.25 (0–0.92)	65 (33.2–88.2)	4.76 (0–9.76)
	*p* value		0.144	0.175	0.053	0.311	0.028 *
Snacking habits	yes	12	214 (181–233)	12.9 (8.8–18.5)	0.59 (0–1.31)	72.4 (56.3–88.6)	4.88 (0–16.3)
	no	24	187 (168–213)	8.5 (2.6–14.0)	0 (0–0.45)	43.9 (22.9–70.2)	0 (0–4.76)
	*p* value		0.093	0.07	0.038 *	0.029 *	0.056
Drinking frequency	≥once during the study	11	180 (172–187)	5.24 (2.3–11.4)	0 (0–0)	40.9 (23.8–63.2)	0 (0–0)
	None during the study	25	205 (176–220)	11.7 (5.3–17.7)	0.25 (0–0.92)	65 (33.2–77.2)	4.76 (0–9.76)
	*p* value		0.144	0.175	0.053	0.311	0.028 *
Snacking frequency	≥once a day	8	226 (207–247)	13.4 (8.9–21.2)	0.7 (0.3–2.82)	77.2 (57.6–94.2)	8.1 (4.45–33)
	<once a day	28	184 (168–211)	9.5 (2.6–15.0)	0 (0–0.52)	48.6 (22.9–70.2)	0 (0–4.76)
	*p* value		0.003 *	0.048 *	0.005 *	0.007 *	0.005 *
Average daily step counts	≥the median (6968 steps per day)	18	187 (159–216)	8 (1.6–14.2)	0 (0–0.74)	52.2 (18.9–69)	0 (0–5.83)
	<the median	17	206 (175–224)	11.3 (5.1–16.4)	0.3 (0–0.81)	65.7 (35.6–90.5)	4.55 (0–9.64)
	*p* value		0.241	0.156	0.331	0.156	0.349
Maximal daily step counts	≥the median (11,937 steps per day)	18	196 (177–225)	9.7 (3.6–15.9)	0.13 (0–1)	59.1 (22–74)	2.17 (0–10.1)
	<the median	17	187 (171–214)	10.9 (3.7–15.1)	0 (0–0.51)	52.4 (33.8–84.4)	0 (0–5.16)
	*p* value		0.447	0.921	0.46	0.78	0.517
Minimal daily step counts	≥the median (2681 steps per day)	18	181 (164–195)	6.4 (1.9–12.4)	0 (0–0.14)	41.9 (20.8–65.4)	0 (0–1.09)
	<the median	17	212 (191–228)	12.2 (6.7–19.9)	0.46 (0–0.92)	71.4 (43.9–84.4)	4.76 (0–10.3)
	*p* value		0.026 *	0.027 *	0.02 *	0.056	0.01 *

*p* values were obtained from a Mann–Whitney U test used to compare the two categories. Participants were stratified by whether they were taking a food item equal to or more than once during the study period (drinking) or once a day (snacking); if they skipped breakfast or had late dinner at least once during the study period; or if they had drinking or snacking habits identified by the self-reported survey questionnaire. For the analysis of daily walking step count, participants were stratified by median split into upper and lower categories of maximal, average, and minimal step count during the study. IQR, interquartile range; CGM, continuous glucose monitoring; CGM max, the maximal sensor glucose level during CGM; TAR, time above range; TAR > 140 and 200, the percentages of time above sensor glucose 140 and 200 mg/dL, respectively; % of ≥140 and 200 peak per meal, proportions of postprandial hyperglycemia equal to or exceed 140 and 200 mg/dL, respectively. In the present study, we defined snacking as ingestion of any food other than meals and drinking as ingestion of alcohol. *, *p* value < 0.05.

**Table 4 nutrients-13-03092-t004:** Associations between the categories of disposition index, snacking frequency, or minimal steps and the risk of having the maximal CGM glucose level ≥200 mg/dL.

Univariate Analysis	Model 1			Model 2			Model 3		
	Odds	95% CI	*p* value	Odds	95% CI	*p* value	Odds	95% CI	*p* value
Disposition index ≤1.57	12.6	1.4–117.6	0.007 *	-	-	-	-	-	-
Snacking frequency ≥once per day	-	-	-	12.6	1.4–117.6	0.007 *	-	-	-
Minimal step category ≤2499	-	-	-	-	-	-	8.4	1.8–38.6	0.003 *
**Bivariate Analysis**	**Model 4**			**Model 5**			**Model 6**		
	Odds	95% CI	*p* value	Odds	95% CI	*p* value	Odds	95% CI	*p* value
Disposition index ≤1.57	12.3	1.6–262.9	0.014 *	-	-	-	11.1	1.3–247.0	0.024 *
Snacking frequency ≥once per day	12.3	1.6–262.9	0.014 *	11.1	1.3–247.0	0.024 *	-	-	-
Minimal step category ≤2499	-	-	-	7	1.4–42.4	0.016 *	7	1.4–42.4	0.016 *
**Trivariate Analysis**		**Model 7**							
	Odds	95% CI	*p* value						
Disposition index ≤1.57	14.5	1.4–376.4	0.022 *						
Snacking frequency ≥once per day	14.5	1.4–376.4	0.022 *						
Minimal step category ≤2499	6.6	1.1–54.7	0.036 *						

Logistic regression models were constructed to examine the risk of having the maximal CGM sensor glucose level ≥200 mg/dL, including the categories of the disposition index, snacking times per day, and minimal daily walking step as predictor variables. Cutoff values were determined by receiver-operating characteristic curves for detecting the maximal sensor glucose level during CGM ≥ 200 mg/dL. Numbers of variables included in the analyses were 1 (univariate, model 1–3), 2 (bivariate, model 4–6), and 3 (trivariate, model 7). CI, confidence interval; CGM, continuous glucose monitoring. In the present study, we defined snacking as ingestion of any food other than meals. *, *p* value <0.05.

**Table 5 nutrients-13-03092-t005:** Characteristics of the study participants according to the snacking habits category.

Parameters	Snacking Habits Category						*p* Value
	Snacking Habits (+) (*n* = 12)			Snacking Habits (−) (*n* = 24)			
	Median	IQR, Lower	IQR, Upper	Median	IQR, Lower	IQR, Upper	
Age, years	54.0	50.5	56.3	56.0	52.3	58.0	0.187
BMI, kg/m^2^	27.7	26.3	31.8	27.9	26.5	29.2	0.737
HbA1c, %	5.5	5.3	5.9	5.3	5.1	5.5	0.039 *
1,5-AG, μg/mL	19.6	11.8	26.5	20.2	15.4	24.1	0.801
HOMA-β	142.0	106.6	277.5	87.9	61.5	128.8	0.002 *
HOMA-IR	2.6	2.1	4.2	1.6	1	2.3	0.001 *
Insulinogenic index	0.9	0.4	1.6	0.6	0.3	1.0	0.46
Matsuda index	2.7	1.3	3.8	4.9	3.0	7.3	0.006 *
Disposition index	1.5	1.4	4.4	2.9	2.0	4.7	0.159
QUICKI	0.33	0.31	0.34	0.36	0.34	0.38	0.001 *
OGTT PG 0, mg/dL	90.5	84.8	98.3	92.5	86.8	96.8	0.724
OGTT PG 30, mg/dL	167.5	134	192.3	153.0	137.3	175.3	0.46
OGTT PG 60, mg/dL	170.5	162.0	218.3	179.0	146.0	193.0	0.557
OGTT PG 120, mg/dL	125.0	95.3	159.3	110.5	95.5	127.8	0.261
OGTT IRI 0, μU/mL	11.5	9.4	22.8	6.8	5.0	9.7	0.001 *
OGTT IRI 30, μU/mL	61.5	39.9	110.6	49.3	29.6	73.6	0.202
OGTT IRI 60, μU/mL	97.8	56.4	188.2	55.5	43.0	132	0.093
OGTT IRI 120, μU/mL	116	39.1	180.6	41.0	24.7	66.9	0.017 *

Data are medians (IQR, interquartile range). *p* values were obtained from a Mann–Whitney U test used to compare the snacking habits categories. BMI, body mass index; 1,5-AG, 1,5-anhydroglucitol; HOMA, homeostatic model assessment; QUICKI, quantitative insulin sensitivity check index; OGTT, 75 g oral glucose tolerance test; PG 0, 30, 60, and 120 pre-load and 30 min, 60 min, and 120 min post-load plasma glucose levels, respectively; IRI 0, 30, 60, and 120 pre-load and 30 min, 60 min, and 120 min post-load serum insulin levels, respectively. In the present study, we defined snacking as ingestion of any food other than meals. *, *p* value < 0.05.

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
