# Peer review of "Impact of Lifestyle Behaviors on Postprandial Hyperglycemia during Continuous Glucose Monitoring in Adult Males with Overweight/Obesity but without Diabetes"

_nutrients, 2021, doi:10.3390/nu13093092_

Round 1

Reviewer 1 Report

This paper includes findings from a study conducted to characterize lifestyle behaviors associated with glucose levels in a sample of nondiabetic men with overweight and obesity. The idea of using continuous glucose monitoring to identify behaviors associated with elevated glucose levels is novel and the research has the potential to offer new insights. These strengths notwithstanding, there are several concerns related to the selected approach and description of study methodology that require further justification and clarification. These are detailed in the following comments.

(1) The Introduction would benefit from clear rationale regarding the selection of the specific study sample - i.e. men between the ages of 50-65 years with overweight/obesity. It is not clear whether the same findings would emerge in women, which should be noted as a study limitation.

(2) The authors cite a previously published paper as a reference for describing characteristics of study participants. Additional information regarding the sample should be provided in the current paper to assist the reader. Further, it is important to clarify how the current manuscript is distinct from findings reported in the previously published work.

(3) It would be helpful to organize the Methods section such that it is divided into clear subheadings related to Participants, Measures, Procedures, and Statistical Plan. Along with this, expanded information is needed regarding the specific lifestyle behaviors that were assessed. Details are needed regarding how participants recorded snacks, step counts, etc. There is relatively more  information regarding assessment of glucose monitoring. In addition to requiring details related to how lifestyle behaviors were assessed, it would be helpful to include a rationale for why these particular lifestyle behaviors were selected for inclusion in the study.

(4) With regard to participant characteristics, please clarify whether the absence of dysglycemia and values on a 2-hour glucose tolerance test were specific inclusion criteria, or whether this was only required for people who happened to be tested (page 2, lines 57-60). 

(5) Without explicit description of the measures included in the study, it is difficult to follow information regarding stratification of participants according to drink, snacking, etc. (page 2, lines 86-89).

(6) The analytic approach requires further justification. In particular, many of the lifestyle variables were examined categorically rather than continuously, with rank order correlations conducted. A rationale for this approach and selection of specific cutoff values is needed. It would also be helpful to include some description of study power and consideration of correction for multiple tests.

(7) As noted above, the Discussion section should include a description of study limitations - e.g. homogeneous study sample, lifestyle behaviors are self-reported, etc. 

(8) The title of the manuscript should be revised to reflect the fact that the authors examination of lifestyle behaviors was broader than snacking.

(9) The manuscript needs to be reviewed and updated with person first language - i.e. adult males with overweight/obesity.

Reviewer 2 Report

The paper is interesting but some more detail is needed in methods, results and discussion.

Line 79 details of study participants have been outlined before but it is difficult to interpret this paper without acknowledging the participants anthropometry in the methods for example i.e. BMI. 

It is interesting so much is made of the contribution of snacks and yet the 75th percentile of intake is only 0.86 per day. What is the explanation of drinking frequency and what beverages are we talking about as some may have differential effects on glucose e.g. tea versus fruit juice? Do you mean alcohol???

I did not find the Table 3 heading gave me sufficient explanation for the table to be interpreted alone as the measure of median and IQR are not included. I could not tell if the text immediately after the tables was meant to be a footnote as typed.

The discussion does not speak of the limitations of the study - they are all men and only 36 people. Would this apply to women? While the glucose monitoring provides fine grain data the estimate of physical activity and diet are very coarse grain data ie how do we know they did not skip photographing meals and snacks. No mention is made of how you differentiated between meals and snacks. Was it timing, foods or types of foods?  Steps must influence glucose differently according to intensity?  More detail that acknowledges some quite serious limitations of these measurements must be made.
